# In Vivo Models of HDV Infection: Is Humanizing NTCP Enough?

**DOI:** 10.3390/v13040588

**Published:** 2021-03-31

**Authors:** Katja Giersch, Maura Dandri

**Affiliations:** 1Department of Internal Medicine, University Medical Center Hamburg-Eppendorf, 20246 Hamburg, Germany; 2German Center for Infection Research (DZIF), Hamburg-Lübeck-Borstel-Riems Site, Germany

**Keywords:** mouse model, infection, hepatitis delta, NTCP, human liver chimeric mice, HDV persistence, HDV replication, host restriction factors, innate immunity, chronic viral hepatitis

## Abstract

The discovery of sodium taurocholate co-transporting polypeptide (NTCP) as a hepatitis B (HBV) and delta virus (HDV) entry receptor has encouraged the development of new animal models of infection. This review provides an overview of the different in vivo models that are currently available to study HDV either in the absence or presence of HBV. By presenting new advances and remaining drawbacks, we will discuss human host factors which, in addition to NTCP, need to be investigated or identified to enable a persistent HDV infection in murine hepatocytes. Detailed knowledge on species-specific factors involved in HDV persistence also shall contribute to the development of therapeutic strategies.

## 1. Hepatitis Delta Virus

The hepatitis delta virus (HDV) was discovered in Italy in 1977 [1] and still causes at least 15–20 million chronic infections worldwide [2]. Recent metanalyses estimated the HDV infection prevalence even at 62–72 million [2,3,4]. HDV is not only the smallest RNA pathogen known to interact with a human host, but it is also a satellite virus which needs the expression of the envelope proteins of the hepatitis B virus (HBV) for the release of HDV particles and propagation among hepatocytes. Liver disease associated with chronic hepatitis D (CHD) causes substantial global morbidity (liver cirrhosis, hepatocellular carcinoma) and mortality [5,6]. The nucleocapsid-like ribonucleoprotein complex (RNP) of the virus contains an only 1679 nucleotide-long, negative-stranded, circular RNA (genomic RNA) and around 200 molecules of hepatitis delta antigen (HDAg) which is the only known protein encoded by the HDV RNA [7,8]. The genomic HDV RNA is single-stranded but folds into an unbranched, rod-like structure with many paired nucleotides [9]. HDV RNA replication occurs in the nucleus of hepatocytes by hijacking mainly the RNA host polymerase II (RNA polymerase I and III are also discussed), which amplifies the genomic HDV RNA through its complementary antigenomic HDV RNA (double rolling circle) [10]. Additionally, a smaller linear HDV mRNA is produced during HDV propagation and encodes for the HDAg which exists in a small and large form. The small HDAg is required for virus replication, while the large variant inhibits replication and promotes virion assembly in the endoplasmatic reticulum [11,12]. The large HDAg is generated during virus replication by post-transcriptional RNA editing at adenosine 1012 (amber/W site) which is mediated by the RNA-specific adenosine deaminase (ADAR) [13]. The balance between viral replication and assembly is orchestrated by the ratio of small and large HDAg and by different post-translational modifications of these proteins, such as prenylation, phosphorylation, methylation, and sumoylation [14]. The HDV RNP (composed of HDV genome and HDAg) is packaged by three HBV envelope proteins termed small (S), medium (pre-S2), and large (pre-S1) hepatitis B surface antigen (HBsAg) and host cell membranes [15]. Therefore, HDV infections occur either upon simultaneous co-infection with HBV or as a super-infection in patients already infected with HBV [16]. Interestingly, the envelope proteins of HBV-unrelated viruses (e.g., dengue virus, hepatitis C virus (HCV), or west Nile virus) were recently discovered to act as alternative helper viruses enabling coating of HDV in vitro [17]. However, clinical relevance appears unlikely, since three different screenings of HCV-infected patients revealed only one case with detectable HDV RNA in the absence of HBV [18,19,20]. In specific clinical cases, HDV was shown to persist in the presence of very low levels of HBV for years [21,22] and HDAg-positive cells have been detected up to 19 months after liver transplantation without any evidence of HBV replication [23,24].

HDV is classified into eight different genotypes with a sequence variation from 19–38% [25,26]. The most common genotype 1 is detected worldwide, including Europe, North America, the Middle East, North Africa, India, and partly in South America and Asia. Genotypes 2 and 4 are mainly observed in the Far East, whereas genotype 3 is restricted to South America and is associated with a more severe disease outcome [27,28]. The genotypes 5, 6, 7, and 8 are almost exclusively found in Africa or African migrants [29].

Due to its simplicity in structure and the lack of its own polymerase, HDV offers much fewer therapeutic targets than other viruses and common HBV therapies based on the use of nucleos(t)ide analogues (NAs) inhibiting the HBV polymerase cannot directly target HDV infection. In 2020, the HBV/HDV entry inhibitor Hepcludex (bulevirtide/Myrcludex-B) obtained conditional marketing authorization as the first HDV-specific drug, while pegylated interferon alpha has commonly been used as an off-label treatment against HDV for decades. Pegylated interferon alpha is associated with severe side effects and weekly treatment only leads to a sustained virological response (defined as undetectable serum HDV RNA six months after treatment) in about 25–30% of the patients with the occurrence of high relapse rates after treatment cessation [30,31,32,33,34,35]. Hepcludex is well tolerated and reduces serum HDV RNA levels significantly after 48-week treatment [36]. Synergistic effects are observed in the combination with pegylated interferon alpha resulting in undetectable HDV RNA viremia in 40% of treated patients after six months of follow up [37]. Pegylated interferon lambda and the prenylation inhibitor lonafarnib (which blocks the interaction between large HDAg and HBsAg during HDV assembly) are currently tested in clinical trials [38].

## 2. Natural Hosts of HDV and HDV-Related Viruses

HDV shows a strict hepatotropism and infects human hepatocytes by using the sodium taurocholate co-transporting polypeptide (NTCP), a transmembrane transporter for bile acids predominantly expressed in the liver [39]. After a reversible attachment of HDV to cell surface-associated heparin sulfate proteoglycans [40], the large HBsAg/pre-S1 (with its myristoylated N-terminus) irreversibly binds to NTCP [41] and HDV enters the hepatocyte. HDV is uncoated and a nuclear localization signal (NLS) in the HDAg sequence mediates the transport of the HDV nucleocapsid into the nucleus in order to initiate genome replication [42]. Besides humans, HDV and HBV are able to naturally co-infect chimpanzees (*pan troglodytes*) [15] and tree shrews (*tupaia belangeri sinensis*) [43], which are small non-rodent animals from Southeast Asia phylogenetically close to primates. Experiments with chimpanzees revealed important information about the establishment of HDV infection and viral persistence [44,45,46], but in 2011, the National Institutes of Health (NIH) placed a temporary moratorium on new studies using chimpanzees due to ethical concerns. Other species closely related to humans, such as Old World monkey baboons, New World monkey tamarins, and crab-eating monkeys (*macaca fascicularis*), do not support HBV and HDV binding and infection due to differences in their NTCP protein sequence [39,47]. By transducing primary hepatocytes from different species with adeno-associated viral vectors encoding hNTCP, mouse, rat, dog, pig, crab-eating/cynomologus macaque (*macaca fascicularis*) and rhesus macaque (*macaca mulatta*) hepatocytes permitted establishment of HDV infection [48]. Interestingly, hepatocytes from pig, crab-eating/cynomolgus macaque (*macaca fascicularis*), and rhesus macaque (*macaca mulatta*) became even fully susceptible to HBV upon hNTCP expression with efficiencies comparable to human hepatocytes [48]. The amino acids 157 to 165 of NTCP were found to be critical for viral entry and viral infection in other species since their substitution with the corresponding human NTCP residues converted crab-eating/cynomolgus macaque (*macaca fascicularis*) NTCP (which shares a 96.3% protein identity with human NTCP) into a functional receptor for pre-S1 binding and allowed HDV infection in vitro [39,49]. Woodchucks (*marmota monax*), ground and tree squirrels (*spermophilus beecheyi*), and pekin ducks (*anas domesticus*) can be infected with human HDV enveloped by HBV-related hepadnavirus proteins and can therefore be used as surrogate animal models for the study of HBV/HDV co-infections [50,51]. The woodchuck model has mainly been used to investigate acute and chronic HDV super-infections [52,53,54] as well as HDV immunization strategies [55,56,57]. However, their limited availability, high costs, genetic diversity as outbred animals, and difficulties in the experimental performance of these hibernating relatively large animals restrict their use for further HDV infection studies.

Interestingly, HDV-like agents which share genomic characteristics with human HDV (circular RNA, self-complementary, unbranched rod-like structure, around 1,700 nucleotides in length, encoding for proteins similar to human HDAg), were recently found in rodents (*proechimys semispinosus*) [58], snakes (*boa constrictor* and *liasis mackloti savuensis*) [59,60], birds [61], fish (classes *actinopterygii, chondrichthyes,* and *agnatha*), amphibians (*bufo gargarizans* and *cynops orientalis*), and invertebrates (*schedorhinotermes intermedius*) [62]. These HDV-like viruses were detected in different organs (brain, liver, kidney, gut, lung) and persisted completely in the absence of a co-infection with HBV-related hepadnaviruses. A recent study performing a transcriptome data analysis also identified novel HDV-related sequences in birds and mammals including the zebra finch (*taeniopygia guttata*), common canary (*serinus canaria*), Gouldian finch (*erythrura gouldiae*), Eastern woodchuck (*marmota monax*), and white-tailed deer (*odocoileus virginianus*) [63]. Moreover, snake HDV was able to replicate in different snake, monkey, and human cell lines after their transfection with a plasmid expressing snake HDV genomes. Infectious snake HDV particles could be produced in vitro when stable snake HDV infected cultured cells were super-infected with HBV-unrelated snake reptarenaviruses and hartmaniviruses [60]. These findings imply that other viruses are also able to provide envelope proteins for HDV-like viruses and that HDV in these species is not restricted to its tissue tropism and might cause diseases other than hepatitis infections [60].

## 3. In Vivo Models to Study HDV Infection

Since HDV infection remains a difficult-to-treat disease with a substantial global health burden, small animal models are needed to investigate the complete viral life cycle, virus host interactions, and the efficacy of antiviral treatments.

### 3.1. Early HDV Experiments in Mice

In 1993, HDV obtained from an experimentally infected woodchuck was injected into the tail vain or peritoneal cavity of mice and reached the liver five to ten days after inoculation. Genomic and antigenomic HDV RNA forms accumulated in mouse livers, leading to the detection of few HDAg positive hepatocytes (less than 0.6%) before virus clearance occurred within 20–30 days [64]. Polo et al., intramuscularly inoculated plasmid DNAs containing head-to-tail cDNA dimers of HDV into mice and detected genomic and antigenomic HDV RNA for up to seven weeks after virus inoculation. Although no HDV mRNA could be determined (by Northern Blot analysis), murine myocytes appeared positive for HDAg [65].

### 3.2. Transgenic Mouse Models

To overcome the narrow host range of HDV and establish convenient mouse models, researchers attempted to generate transgenic mice as a model of HDV replication. Guilhot et al., developed transgenic mice producing the small and large form of HDAg in their hepatocytes. During 18 months of observation, no biological or histopathological evidence of liver disease was detectable [66]. In another attempt, transgenic mice were generated which expressed replication-competent genomic dimers of HDV RNA. Although these transgenic mice were able to replicate RNA transcripts in different tissues, intrahepatic genome replication occurred only in a very limited number of hepatocytes (less than 1%); RNA-editing to produce the large HDAg did not take place and mRNA species could not be detected. Similarly, signs of liver cytotoxicity were missing completely, suggesting that transgenic mice cannot be used as a suitable model for the study of chronic HDV replication [67].

### 3.3. Hydrodynamic Mouse Models

In 1999, the hydrodynamic-based transfection model emerged, creating new possibilities to generate rodents harboring hepatotropic viruses. By rapidly injecting DNA in a large volume into the tail vein of mice, organs, especially the liver, start to express transgenes that reach a peak after hours and decrease after days to weeks [68]. Chang et al. injected HDV cDNA or in vitro transcribed HDV RNA into mice via hydrodynamic transfection. Five to nine days after inoculation, an increasing accumulation of genomic HDV RNA was detected intrahepatically, which then decreased from day 15 to 30 [69]. This model might help studying aspects of genome replication and RNA transcription in the context of acute HDV infection. However, the virus uptake is not based on a natural receptor-mediated infection. Murine host immune responses eliminate the virus particles after a few days and the rapid injection of large volumes of fluids also causes liver damage, thus limiting studies focusing on HDV-associated liver disease [70].

### 3.4. Human Liver Chimeric Mice

Rodents cannot be infected with HDV and HBV but during the past 20 years, different strategies were explored to augment engraftment and transplantation efficiency of human hepatocytes into mice in order to develop in vivo systems enabling infection studies with HDV and HBV. It is well known that the liver is a highly regenerative organ and after liver injury or surgical removal of more than two thirds of the liver, its mass can be restored completely. Therefore, the idea of transplanting cells either ectopically or into the liver to restore its structure and function emerged. Investigations first showed that isolated primary human hepatocytes were able to contribute to liver function after being ectopically transplanted into dorsal fat pads [71], spleen [72], peritoneal cavity [73], or under the renal capsule of immune deficient mice [74].

In 2000, Ohashi et al., developed a novel small animal model for the study of HBV mono- and HDV super-infections. Isolated human hepatocytes were engrafted via the kidney capsule into livers of non-obese diabetic (NOD) mice crossbred with severe combined immunodeficiency (SCID) mice. SCID mice lack T- and B-lymphocytes due to a spontaneous mutation in the Prkdc gene and therefore do not have an adaptive immune system [75]. After human hepatocytes integrated into mouse livers without losing their characteristic morphology and phenotype, inoculation with HBV and super-infection with HDV was performed successfully, thus proving the feasibility of generating human chimeric mice that supported the full life cycle of HDV and HBV [75]. This model demonstrated that human hepatocytes can survive in immunodeficient mice for months while the level of host liver repopulation was very low.

Alb-uPA-transgenic mice, first developed in 1990 to study neonatal bleeding disorders [76], express the hepatotoxic urokinase-type plasminogen activator (uPA) transgene under the control of an albumin promoter causing subacute murine hepatocyte toxicity soon after birth, thus providing a growth advantage for transplanted human hepatocytes. In 2001, Dandri et al. transplanted human hepatocytes isolated from an adult human liver into uPA-transgenic mice which were backcrossed with immunodeficient B and T cell lacking Rag-2^-/-^ (recombinant activation gene-2) mice, demonstrating successful intrahepatic engraftment of human hepatocytes and reconstitution of mouse livers up to 15% [77]. Transplanted human hepatocytes remained permissive for human hepatotropic viruses which was proven by the establishment of a productive HBV infection after inoculating uPA/Rag-2^-/-^ mice with a human HBV-DNA-positive serum [77]. By using Alb-uPA homozygous instead of heterozygous, mice repopulation rates could be increased from 15% [77] to more than 50%, and therefore augmenting the chance of establishing an extensive virus infection and facilitating investigation of virus-hepatocyte interactions [78]. Hepatocytes from a human donor were also engrafted into livers of immunodeficient uPA mice which were crossed with SCID or SCID/beige mice (lacking additionally natural killer (NK) cells) and were successfully infected with HBV or HCV [78,79,80]. In 2012, our group infected human liver chimeric uPA/SCID/beige mice (short USB mice) or uPA/SCID/Il2rg^-/-^ mice (short USG mice) which lack B, T, and NK cells with a human HDV-RNA-positive serum [81] and for the first time created an efficient mouse model for the study of HBV and HDV infections [81,82,83], virus-host interactions [84], and preclinical drug evaluation [81,85,86]. In human-liver chimeric mice, infection and replication of HBV and HDV lasts for several months or to the life span of the animal. Of note, these studies showed that HBV/HDV co-infection is associated with a strong induction of human type I interferons compared to HBV mono-infection [84]. Moreover, we showed that HDV mono-infection can persist also in the absence of HBV for several weeks [82] and during liver regeneration by being amplified through cell division [83]. To date, this model was also used to evaluate anti-HDV treatments including the antiviral activity of pegylated interferon alpha and lambda [85], the efficacy of the HBV/HDV entry inhibitor Myrcludex-B to hinder HDV de novo infection [81], and the outcome of the HBV-targeting siRNA ARB-1740 intervention against HDV [86].

Another human-liver chimeric model where efficient liver repopulation can be achieved is the fumarylacetoacetate hydrolase (fah)-deficient mouse model. Fah is the last enzyme of the tyrosine catabolism cascade and its deletion in fah-deficient mice leads to an accumulation of hepatotoxic intermediates of tyrosine catabolism (e.g., fumarylacetoacetate) which injure host hepatocytes [87]. Accumulation of fumarylacetoacetate can be prevented by the pharmacologic inhibitor of tyrosine catabolism 2-(2-nitro-4-trifluormethylbenzoyl)-cyclohexane-1,3-dione (NTBC), and cyclic administration of NTBC allows control of liver failure in Fah^-/-^ mice. Since 2010, Fah^-/-^ mice have been backcrossed with immunodeficient Rag-2^-/-^ and/or Il2rg^-/-^ mice (shortly named FRG mice) and successfully infected with HBV [88,89], while HDV infections were not studied in these mice to date. Limitations of human-liver chimeric mice include relatively high costs, challenging breeding (compared to wildtype mice), hepatocyte donor-to-donor differences, and the lack of an adaptive immune system. While the liver damage can be more controlled in FRG mice (by using NTCB) and animals are generally healthier than USB or USG mice, they often develop tyrosinemia and liver carcinomas [89].

In the last years, several groups attempted to engraft human hepatocytes and human immune cells into uPA mice to generate dually humanized mice and to study hepatitis virus infections in the presence of the adaptive immune system [90,91,92,93,94]. Although significant progress has been made in establishing liver and hematopoiesis double-humanized mice, infection studies were only performed with HBV and not HDV, and this model is still limited by difficulties in generation, high costs, and inefficient immune response upon virus infection [89]. Recently, HBV/HDV infected human-liver chimeric mice reconstituted with HLA-A0201 hepatocytes were challenged with haplotype matched T cell receptor (TCR)-redirected T cells engineered from T cells of a healthy individual to recognized HBV infected hepatocytes [95]. While this study highlighted the immunogenic properties of HDV and revealed the therapeutic potential of using HBV-specific engineered T cells for HDV treatment, this approach also demonstrated that uPA/SCID/Il2rg^-/-^ mice can successfully be reconstituted with human hepatocytes and human T cells [95].

### 3.5. AAV-Based Mouse Models

In 2017, Suárez-Amarán et al., and Usai et al., established an HDV replication model based on the use of adeno-associated vectors (AAVs) to deliver HDV and HBV replication-competent genomes [96,97], or HDAgs [97], into immunocompetent C57BL/7 wildtype mice. HDV replication and HDV genome editing were observed for 45 days although the HDV infection clearly decreased already after 21 days [96,97] and the serum of mice at time points later than 14 days post HDV infection was not infectious in vitro [96]. HBV/HDV co-infection in this model was associated with the reduction of HBV antigen expression, induction of type I interferon responses and the onset of liver damage. In this model, the mitochondrial antiviral-signaling protein (MAVS), which is known to be essential for antiviral innate immunity, was identified as a main player in HDV detection [96], and tumor necrosis factor alpha (TNFα) was shown to play a major role in the development of HDV-induced liver damage which could be blocked by treatment with the TNFα inhibitor eternacept [97]. Overall, the AAV-based HDV replication model mimics acute severe HDV infection with liver inflammation and injury and permits investigation of HDV-mediated liver damage and assessment of new treatments. However, this model cannot be applied for the study of HDV entry or the efficacy of novel entry inhibitors. Moreover, potential species-related differences at the level of intrinsic, innate responses of the target cells, the hepatocytes, cannot be ruled out when studying HDV replication in murine hepatocytes [98].

### 3.6. NTCP-Based Mouse Models

The discovery of NTCP as the bona fide entry receptor of HBV and HDV raised new possibilities to establish immunocompetent HDV infection mouse models [39]. Previous studies showed that although the HBV and HDV entry inhibitor Myrcludex-B and pre-S1 are able to specifically bind to mouse hepatocytes [99,100], murine NTCP does not support HBV and HDV infection and replication [49]. In 2015, neonatal (but not adult) C57BL/6 transgenic mice exogenously expressing human NTCP (hNTCP) were shown to support HDV infection and replication as evidenced by the presence of antigenomic RNA and edited RNA species responsible for the translation of the large delta antigen in the livers of infected mice [101]. Approximately 3% of mouse hepatocytes were determined to be HDV positive six days post infection. However, clearance of HDV infection occurred already within 12 days and appeared independent to adaptive immune responses since rapid resolution of HDV was also observed in hNTCP/SCID mice lacking B and T cells [101]. Intriguingly, mice older than four weeks could not be infected with HDV in that system, suggesting that the susceptibility of HDV in murine hepatocytes may be not only age-dependent but it could also be limited by the maturation of the immune system [89,101]. The production of pro-inflammatory cytokines is reduced in newborns and the dendritic cells system required for adaptive immunity in mice is not fully developed until five weeks of age, which renders newborns at risk for viral infections [101,102,103,104]. Winer et al. generated immunocompetent mice transgenically expressing a bacterial artificial chromosome (BAC) containing part of human chromosome 14, which includes the SLC10A1 gene encoding NTCP [105]. These so-called hNTCP/BAC mice were infected with HDV in the presence or absence of a 1.3x HBV transgene (which leads to sustained secretion of HBsAg), became transiently viremic, and supported HDV replication for 30 and 14 days, respectively. Immunodeficient, non-obese diabetic (NOD)/recombinase activating gene 1 knockout (Rag1^−/−^)/interleukin 2 receptor gamma chain null (IL2rg^−/−^) (short NRG) mice co-expressing both hNTCP and 1.3x HBV transgenes were shown to support HDV infection for at least 80 days [105]. Although this model does not support HBV replication and spreading, it is plausible that the presence of HBV supported not only HDV assembly and release but also occurrence of new infection events, thus contributing to the longer maintenance of HDV markers in the livers of these mice. In this study, hNTCP/BAC-NRG mice were also challenged with Myrcludex-B and lonafarnib [105]. Please note that hNTCP transgenic mice still endogenously express the murine NTCP receptor which could theoretically lead to species-related interactions with hNTCP and influence HDV entry.

HDV infection through murine NTCP (mNTCP) is restricted by only three amino acids [49]. Substituting the residues 84, 86, and 87 of the mNTCP with the human counterparts by TALEN or CRISPR/Cas technology renders mice susceptible to HDV but not HBV [98,106]. Although these mice were successfully infected with different HDV genotypes and HDV replication and editing was confirmed, the infection efficacy was low (less than 0.1% of hepatocytes were HDAg positive) and HDV was cleared within 21 days and without detectable signs of apoptosis in mouse livers [98]. Strikingly, murine hepatocytes of mice with humanized NTCP also cleared HDV infection within 21 days when they were isolated and transplanted into immunodeficient uPA/SCID/beige mice and without inducing detectable interferon stimulated genes (ISGs) or chemokines in HDV mono-infected cells [98]. In contrast, HDV RNA and HDAg persisted in HDV mono-infected human hepatocytes in the same immunodeficient human-liver chimeric mice for at least six weeks, thus providing evidence that species-specific differences are likely to account for the impaired survival capacity of HDV mono-infection within murine hepatocytes. In line with our observations, also chimpanzee hepatocytes which were isolated and cultured in vitro were shown to support HDV mono-infection for at least 42 days [107]. These findings suggest that not innate or adaptive immune responses but, apart from NTCP, further species-specific characteristics limit HDV infection efficacy and persistence in murine hepatocytes.

Although NTCP-based mouse models are relatively easy to establish and not hampered by donor-to-donor variability, there are still many limitations including the low infection efficacy, the absence of liver damage, and the short duration of HDV infection. Despite humanizing NTCP or using hNTCP transgenic mice, murine hepatocytes appear less prone to HDV infection compared to human livers, suggesting that either murine factors hinder HDV infection endurance or that additional human dependency factors are required to create more efficient immunocompetent murine HDV infection models. Moreover, reasons for the inability of murine hepatocytes to sustain HDV infection efficiently are not understood, but the fast clearance of HDV in murine hepatocytes of immunodeficient mice [98,101,105] and the persistence of HDV in human and chimpanzee hepatocytes [82,107] emphasize the role of species barriers in HDV infection susceptibility and persistence.

Another major disadvantage is that HBV infection could not be established in any of the NTCP-based murine infection models. The absence of HBV strongly contributes to the low infection rates of HDV in these models since HDV only undergoes a single round of infection and cannot spread and infect murine livers further. The reasons for the inability of HBV to establish a productive infection in hNTCP expressing murine cells in vivo [101] and even in vitro [49,108] are unclear. Although HBV and HDV use the same envelope proteins for cell entry, distinct downstream mechanisms are exploited by these viruses to establish infection and therefore a post entry blockade for HBV is conceivable [89]. After binding to NTCP and internalization, HBV is uncoated and transported to the nucleus, where the covalently closed circular DNA (cccDNA) which serves as the transcriptional template for HBV gene expression is formed. The process involved in the formation of the HBV cccDNA minichromosome is currently not fully understood, but it requires the support of multiple cellular factors such as enzymes involved in the DNA repair machinery. While HBV transgenic mice support viral replication but not cccDNA formation [109], transfection of a cccDNA-like molecule into non-susceptible murine cells supported HBV antigen expression, indicating that the main bottle neck for HBV infection in murine cells is the steps preceding cccDNA establishment [110]. The generation of cccDNA requires several host factors including proliferating cell nuclear antigen (PCNA), flap endonuclease 1 (FEN-1), DNA ligase 1 (LIG1), and DNA topoisomerases which might differ between murine and human cells in their capacity to build the cccDNA and support efficient HBV infection [111,112].

## 4. HDV Host Restrictions Factors

Despite the development of several HDV infection/replication models in the past decades, an ideal mouse model which supports the full life cycle of HDV and allows the study of virus-host interactions and potential antiviral therapies in the presence of innate and adaptive immunity still does not exist. The humanization of NTCP in murine hepatocytes or the use of hNTCP-transgenic mice brought us one step closer to the goal of establishing an HDV infection in immunocompetent mice. However, to mimic a chronic and long lasting HDV infection as observed in patients, we likely need to identify and humanize further host restriction factors for HDV as well as to enable substantial intrahepatic HDV spreading.

The epidermal growth factor receptor (EGFR), which is a transmembrane glycoprotein and leads to cell proliferation upon stimulation with epidermal growth factor (EGF), was recently proposed to be a host entry co-factor for HBV and HDV. EGFR was shown to interact with NTCP and to mediate HBV and HDV internalization [113] (Figure 1). Depletion of EGFR in vitro did not affect viral attachment (since cell-surface NTCP expression remained stable) but decreased intracellular levels of HBV DNA and HDV RNA and therefore hindered the establishment of both HBV and HDV infection [113]. Independent of the presence of HBV, NTCP was demonstrated to interact with EGFR by forming a NTCP–EGFR complex which then translocates between the plasma membrane and intracellular vesicles. When NTCP was prevented from interacting with the EGFR (by introducing mutations in NTCP, by masking the binding interface with a decoy peptide, or by inactivating EGFR) the cells no longer supported HBV entry [113]. On the other side, stimulation with EGF was found to potentiate cell susceptibility to HBV infection in vitro [113]. It is known that the sequence of murine and human EGFR are 88% identical and that EGF binding induces phosphorylation of EGFR in both species [114]; however, further studies are needed to assess whether species-related differences in EGFR indeed contribute to the lower infection efficacy in NTCP-based murine hepatocytes, both in vitro and in vivo.

Verrier et al., performed a high-throughput loss-of-function screening and transfection studies in vitro and revealed glypican 5 [115] and the carbamoyl-phosphatesynthetase 2, aspartate transcarbamylase, and dihydroorotase (CAD) enzyme together with the estrogen receptor alpha (or estrogen receptor 1, ESR1) [116] as key host factors for HDV entry and life cycle, respectively (Figure 1).

Glypican 5 belongs to a group of six heparan sulfate proteoglycans (HSPG) which are attached to cell membranes, act as co-receptors, and are known to regulate the signaling activity of growth factors. In this study, glypican 5 was identified as an entry host factor for HBV and HDV through its interaction with the HBV envelope proteins during the initial viral attachment [115]. In vitro experiments showed that silencing of glypican 5 decreased HBV binding to the cell surface and thus HDV and HBV infection in NTCP-expressing human hepatoma cells and primary human hepatocytes [115]. The interaction between HBV envelope proteins and glypican 5 was reported to be independent of the NTCP expression. Although HDV enters murine hepatocytes of mice with human or humanized NTCP through its specific receptor, it is conceivable that HDV attachment to glypican 5 triggers viral entry and that therefore species-related differences of glypican 5 lower the overall efficacy of HDV infection in these models.

CAD is an enzyme playing a key role in the pyrimidine biosynthesis and its expression is regulated by activated ESR1. Silencing of both CAD and ESR1 led to a robust decrease of HDV RNA in infected NTCP-overexpressing Huh7 cells (Huh-106 cells) [116]. Moreover, the authors demonstrated that the pyrimidine/CAD pathway is involved in all the steps of HDV replication including the synthesis of antigenomic RNA, but it is not relevant for HBV replication. Since the pyrimidine biosynthesis leads to the production of de novo nucleotides, this pathway likely serves as a source of providing nucleotides for HDV RNA and is therefore required for optimal virus replication [116]. The role of CAD and ESR1 in HDV replication was corroborated by performing inhibition studies using the specific CAD inhibitor N-(phosphonoacetyl)-L-aspartic acid (PALA) and the selective ESR1 inhibitor fulvestrant (reducing CAD protein expression levels) which both had antiviral effects on HDV in vitro [116]. These results uncover an interesting link between the pyrimidine pathway and HDV replication and provide a promising approach for the development of new antiviral treatments. While it is tempting to speculate that host-specific differences of CAD and ESR1 pathways in mice are responsible for the observed limitations of HDV replication and persistence in NTCP-based mouse models, its relevance in vivo has still to be evaluated.

## 5. Perspective

Despite a vaccination against HBV that protects individuals also from HDV infection and the promising results obtained in the clinical trials investigating the entry inhibitor bulevirtide, HDV infections remain a global health threat with severe clinical courses and unsatisfactory long-term therapy outcomes. To facilitate the development and assessment of new treatment options, not only high-throughput in vitro systems but also in vivo infection models are needed. Such systems must support the full life cycle of HDV and permit investigation of antiviral compounds as well as innate and adaptive immune responses. While the generation of different HDV in vivo models in the past years greatly contributed to study certain aspects of HDV entry, replication, persistence, and virus-host interactions, they all have substantial limitations (Table 1). Chimpanzees allowed the study of HDV in the presence of an infection with HBV and of a complete immune system, but raised strong ethical concerns. HBV/HDV infections can also be studied in immunocompetent AAV-based mouse models leading to liver inflammation and damage. However, HDV persistence seemed to be limited and HDV entry cannot be investigated. Human-liver chimeric mice lack adaptive immune responses and are costly, but they support the full life cycle of HDV and HBV, including efficient HDV spreading and life-long persistence. Lastly, the more recent immunocompetent mice expressing a human or humanized NTCP have shown to be susceptible to HDV infection and permit studies in murine hepatocytes. However, HDV infection efficacy is low, HDV is cleared within three weeks, and HBV infections are not possible. Despite such drawbacks, the latter model revealed that murine hepatocytes are unable to sustain HDV infection per se and that humanization of NTCP is not sufficient to initiate HBV replication, thus highlighting the crucial role of additional species barriers. To establish a persistent HDV infection where the majority of hepatocytes is infected with HDV, spreading mediated by HBV envelope proteins is also indispensable. Future studies aiming at identifying HBV and HDV host restrictions and/or co-factors hindering HDV persistence in murine hepatocytes shall greatly contribute to the development of improved small animal infection models and to the discovery of novel potential host-derived therapeutic targets of HDV.

## Figures and Tables

**Figure 1 viruses-13-00588-f001:**
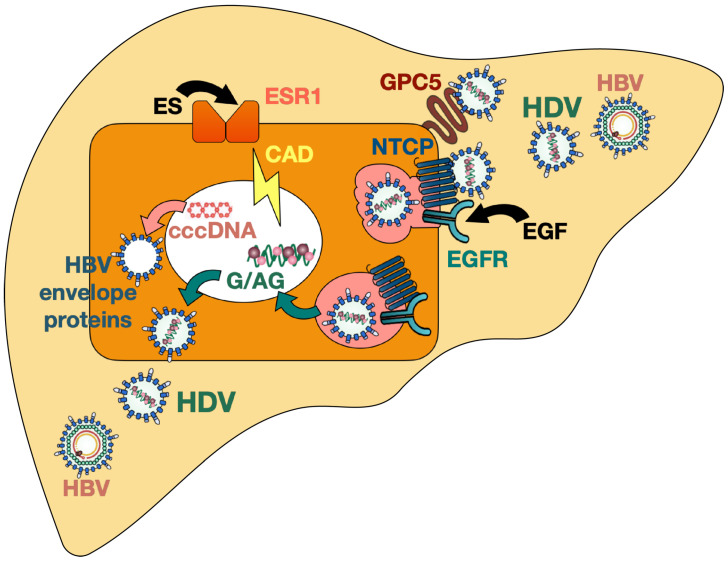
HDV infection and host factors. HDV: hepatitis D virus; HBV: hepatitis B virus; GPC5: glypican 5; NTCP: sodium taurocholate co-transporting polypeptide; EGF(R): epidermal growth factor (receptor); ES(R): estrogen (receptor); CAD: carbamoyl-phosphatesynthetase 2, aspartate transcarbamylase, and dihydroorotase; cccDNA: covalently closed circular DNA; HBsAg: hepatitis B surface antigen.

**Table 1 viruses-13-00588-t001:** Pros and cons of the different HDV in vivo models.

Model	Pros	Cons	Ref.
Chimpanzees	Full HDV life cycle, innate and adaptive immune system, HBV infection	Highly restricted availability, ethical concerns and high costs	[15,44,45,46]
Woodchucks	Full HDV life cycle, innate and adaptive immune system	Co-infection only with woodchuck hepatitis virus (WHV), limited availability, high costs and difficulties in the experimental performance	[50,51,52,53,54,55,56,57]
Hydrodynamic mouse models	Innate and adaptive immune system	No HDV entry, HDV clearance within 30 days, liver damage due to large injection volume	[68,69]
Human liver chimeric mice	Full HDV life cycle, innate immune system, HBV infection, life-long HDV persistence	No adaptive immune system, limited availability, high costs	[75,81,82,83,84,85,86]
AAV based mouse models	Innate and adaptive immune system, liver inflammation and damage, HBV replication	No HDV entry, HDV decline after 21 days until 45 days (long-term data missing)	[96,97]
hNTCP transgenic mice hNTCP/BAC mice	Innate and adaptive immune system	Low HDV infection efficacy, HDV clearance within 21 days, no HBV infection, presence of mNTCP	[101,105]
Mice with humanized NTCP (CRISPR/Cas, TALENs)	Innate and adaptive immune system	Low HDV infection efficacy, HDV clearance within 21 days, no HBV infection	[98,106]

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
