# Peer review of "In Vivo Models of HDV Infection: Is Humanizing NTCP Enough?"

_viruses, 2021, doi:10.3390/v13040588_

Round 1

Reviewer 1 Report

In the present manuscript, Giersch et al summarize in vivo models for establishing HDV expression and discuss the human host factor required for HDV infection. It mainly focuses on the mice model for HDV research. Before the discovery of NTCP as HBV/HDV receptor, most HDV mice model relies on different kinds of HDV delivery method other than infection. The discovery of NTCP opens a new gate for the establishment of HDV infection in mice model. However, murine NTCP does not support HBV/HDV infection. Thus, various human NTCP transgenic mice models are developed. However, the HDV infection efficiency is still low. As a result, there might be some unknown HDV host restriction factors required for efficient HDV infection. The author also discusses EGFR and CAD as newly discovered key host factors for improving HDV HDV infection. The manuscript was fairly written but requires additional proofreading.

Major points:

The concluding remarks of this manuscript are:

-summarizing pros and cons of current HDV in vivo models

-emphasizing the idea that host factors other than human NTCP are required to support efficient HDV infection in mice model

-This article titled’’ In vivo models of HDV infection: Is humanizing NTCP enough?” However, only two pages are used to illustrate NTCP-based mice model. To echo the title, NTCP-based mice model and its limitation can be more elaborated.

-The restriction factors required for HDV infection could be reviewed and discussed in more details.

-In Figure 1. HDV infection and host factors: NTCP is better to be drawn as a seven to nine transmembrane molecule, while EGFR as a Y-shaped monomer or dimer.

Specific comments

  1. Page 1, line 33: “around 200 molecules of hepatitis Delta antigen” à this information can’t be found in reference 7
  2. Page 2, line 51: “The HDV genome is surrounded by three HBV envelope proteins” à the HDV RNP (ribonucleoprotein, composed of HDV genome and HDAg) is packaged by three HBV envelope proteins
  3. Page 2, line 53: “host lipids” à host cell membranes
  4. Page 2, line 67: “The most common genotype 1 is detected in Europe, North America, Middle East, North Africa, India and partly in South America and Asia.” à the most common genotype 1 is detected worldwide, including Europe, North America, Middle East, North Africa, India and partly in South America and Asia.
  5. Page 2, line 73: “Due to its simplicity in structure and the lack of producing its own polymerase” à Due to its simplicity in structure and the lack of its own polymerase
  6. Page 2, line 77: “in 2020 the HBV/HDV …”à in 2020, the HBV/HDV (this should be also edited in other sentences)
  7. Page 3, line 98: “After a reversible attachment of HDV to cell-surface-associated

heparin sulfate proteoglycans [39] the large HBsAg/pre-S1 (with its

myristoylated N-terminus) irreversibly binds to NTCP [40] and HDV enters the hepatocyte.” à After a reversible attachment of HDV to cell-surface-associated

heparin sulfate proteoglycans [39], the large HBsAg/pre-S1 (with its

myristoylated N-terminus) irreversibly binds to NTCP [40] and HDV enters the hepatocyte.

  1. Page 3, line 101: “a signal in the HDAg …” à an NLS signal in the HDAg …
  2. Page 3, line 103: “Besides humans, … “ à Besides from humans, …
  3. Page 5, line 176: genomic and antigenomic HDV RNA were detected in what specimen?
  4. Page 5, line 178: The amount of mRNA is low (about 0.2% relative to genomic RNA). As a result, mRNA is hard to be detected even in the presence of HDAg.
  5. Page 5, line 181: “To overcome …” à To overcome the narrow host range of HDV and establish convenient mice model, researchers …
  6. Page 5, line 184: “But during …” à During …
  7. Page 5, line 192: “Again” à Similarly
  8. Page 5, line 199: “start to express …” à start to express transgenes that reach a peak after hours and decrease after days to weeks
  9. Page 5, line 202: “Chang et al …” à Chang et al injected HDV cDNA or in vitro transcribed HDV RNA into mice via hydrodynamic transfection. Five to nine days after inoculation, an increasing accumulation of genomic HDV RNA …”
  10. Page 6, line 207: “However, … “ à However, the virus uptake is not based on a natural receptor-mediated infection. Murine …
  11. Page 6, line 238: “however” à while
  12. Page 6, line 243: “starting shortly” à soon
  13. Page 7, line 253: “with a human …” à with a human HBV DNA positive serum
  14. Page 7, line 284: “and therefore …” à , which injure host hepatocytes
  15. Page 8, line 292: “are” à includes
  16. Page 8, line 293: “mice, they” à mice, which
  17. Page 8, line 322: “reduction” à the reduction
  18. Page 9, line 351: “not to be dependent on” à independent to
  19. Page 9, line 370: “lonafarnib treatment” à lonafarnib
  20. Page 10, line 384: “cleared HDV infection also” à also cleared HDV infection
  21. Page 10, line 386: “detectable induction of” à detectable
  22. Page 10, line 391: “to generate” à to be regenerated
  23. Page 10, line 392: “major limitations are” à there are still many limitations including
  24. Page 10, line 396: “a post entry block” à a post entry blockade
  25. Page 10, line 408: “does still” à still does
  26. Page 10, line 411: “However to …“ à However, to mimic a chronic and long-lasting HDV infection as observed in patients, we need to identify and humanize further host restriction factors …
  27. Page 13, line 493: “with” à by

Author Response

Reviewer 1:

Comments and Suggestions for Authors

In the present manuscript, Giersch et al summarize in vivo models for establishing HDV expression and discuss the human host factor required for HDV infection. It mainly focuses on the mice model for HDV research. Before the discovery of NTCP as HBV/HDV receptor, most HDV mice model relies on different kinds of HDV delivery method other than infection. The discovery of NTCP opens a new gate for the establishment of HDV infection in mice model. However, murine NTCP does not support HBV/HDV infection. Thus, various human NTCP transgenic mice models are developed. However, the HDV infection efficiency is still low. As a result, there might be some unknown HDV host restriction factors required for efficient HDV infection. The author also discusses EGFR and CAD as newly discovered key host factors for improving HDV HDV infection. The manuscript was fairly written but requires additional proofreading.

Major points:

The concluding remarks of this manuscript are:

-summarizing pros and cons of current HDV in vivo models

-emphasizing the idea that host factors other than human NTCP are required to support efficient HDV infection in mice model  

-This article titled’’ In vivo models of HDV infection: Is humanizing NTCP enough?” However, only two pages are used to illustrate NTCP-based mice model. To echo the title, NTCP-based mice model and its limitation can be more elaborated.

We agree with the reviewer that the title puts the focus on humanized NTCP models and therefore we added further details regarding NTCP based mouse models and their limitations at page 9-11. In the revised manuscript we mentioned and discussed e.g. why hNTCP transgenic mice older than four weeks cannot be infected with HDV or why murine hepatocytes with hNTCP are not susceptible to HBV (see also reviewer 2). Please note that while humanizing NTCP is the most novel aspect in the field, these models unfortunately do not fulfill yet the expectations of researchers. Therefore, the main title remains in vivo models of HDV infection and we intended to provide an overview of all different models available. We also discussed now pros and cons of the current models in more detail on page 8 and in the section “perspectives” on page 15/6. Together with the slightly revised table 1, we hope to provide now a more comprehensive overview.

Since the current models do not support efficient HDV infection, we also revised the manuscript on page 10-11 to explain more properly why we suggest that host factors are required to establish a more efficient and persistent HDV infection in murine hepatocytes in vivo.

-The restriction factors required for HDV infection could be reviewed and discussed in more details. As suggested by the reviewer we extended the section “host restrictions factors” on page 13-14 (see also revised figure 1) and added glypican 5 as potential host restriction factor for viral entry.

-In Figure 1. HDV infection and host factors: NTCP is better to be drawn as a seven to nine transmembrane molecule, while EGFR as a Y-shaped monomer or dimer. Thank you for this suggestion, we changed the receptors in the revised figure 1.

Specific comments

  1. Page 1, line 33: “around 200 molecules of hepatitis Delta antigen” à this information can’t be found in reference 7 Thank you for pointing that out, we added the correct reference (Gudima et al., J Virol. 2002)
  2. Page 2, line 51: “The HDV genome is surrounded by three HBV envelope proteins” à the HDV RNP (ribonucleoprotein, composed of HDV genome and HDAg) is packaged by three HBV envelope proteins We rephrased the sentence as you suggested (page 2, line 58/59).
  3. Page 2, line 53: “host lipids” à host cell membranes We rephrased the sentence as you suggested (page 2, line 61).
  4. Page 2, line 67: “The most common genotype 1 is detected in Europe, North America, Middle East, North Africa, India and partly in South America and Asia.” à the most common genotype 1 is detected worldwide, including Europe, North America, Middle East, North Africa, India and partly in South America and Asia. We rephrased the sentence as you suggested (page 2, line 75).
  5. Page 2, line 73: “Due to its simplicity in structure and the lack of producing its own polymerase” à Due to its simplicity in structure and the lack of its own polymerase We rephrased the sentence as you suggested (page 2, line 81).
  6. Page 2, line 77: “in 2020 the HBV/HDV …”à in 2020, the HBV/HDV (this should be also edited in other sentences) We corrected this sentence and the other sentences.
  7. Page 3, line 98: “After a reversible attachment of HDV to cell-surface-associated

heparin sulfate proteoglycans [39] the large HBsAg/pre-S1 (with its

myristoylated N-terminus) irreversibly binds to NTCP [40] and HDV enters the hepatocyte.” à After a reversible attachment of HDV to cell-surface-associated

heparin sulfate proteoglycans [39], the large HBsAg/pre-S1 (with its

myristoylated N-terminus) irreversibly binds to NTCP [40] and HDV enters the hepatocyte. We corrected the sentence.

  1. Page 3, line 101: “a signal in the HDAg …” à an NLS signal in the HDAg … We rephrased the sentence as you suggested (page 3, line 109/110).
  2. Page 3, line 103: “Besides humans, … “ à Besides from humans, … We corrected the sentence (page 3, line 112).
  • Page 5, line 176: genomic and antigenomic HDV RNA were detected in what specimen? In mouse livers. We specified this as on page 5 line 180/181.
  • Page 5, line 178: The amount of mRNA is low (about 0.2% relative to genomic RNA). As a result, mRNA is hard to be detected even in the presence of HDAg. You are right. We rephrased the sentence: “Although no HDV mRNA could be determined (by Northern Blot analysis),..” and hope it is not misleading anymore (page 5 line 186/187).
  • Page 5, line 181: “To overcome …” à To overcome the narrow host range of HDV and establish convenient mice model, researchers … We rephrased the sentence as you suggested (page 5, line 190).
  • Page 5, line 184: “But during …” à During … Done, thank you.
  • Page 5, line 192: “Again” à SimilarlyDone, thank you.
  • Page 5, line 199: “start to express …” à start to express transgenes that reach a peak after hours and decrease after days to weeks We rephrased the sentence as you suggested (page 5, line 209).
  • Page 5, line 202: “Chang et al …” à Chang et al injected HDV cDNA or in vitro transcribed HDV RNA into mice via hydrodynamic transfection. Five to nine days after inoculation, an increasing accumulation of genomic HDV RNA …” We corrected the sentence (page 6, line 211/212).
  • Page 6, line 207: “However, … “ à However, the virus uptake is not based on a natural receptor-mediated infection. Murine … We rephrased the sentence as you suggested (page 6, line 216).
  • Page 6, line 238: “however” à while Done, thank you.
  • Page 6, line 243: “starting shortly” à soon Done, thank you.
  • Page 7, line 253: “with a human …” à with a human HBV DNA positive serum Done, thank you.
  • Page 7, line 284: “and therefore …” à , which injure host hepatocytes We rephrased the sentence.
  • Page 8, line 292: “are” à includes Done, thank you.
  • Page 8, line 293: “mice, they” à mice, which We think this sentence is correct, since FRG mice often develop tyrosinemia and liver carcinomas.
  • Page 8, line 322: “reduction” à the reduction Done, thank you.
  • Page 9, line 351: “not to be dependent on” à independent to Done, thank you.
  • Page 9, line 370: “lonafarnib treatment” à lonafarnib Done, thank you.
  • Page 10, line 384: “cleared HDV infection also” à also cleared HDV infection Done, thank you.
  • Page 10, line 386: “detectable induction of” à detectable Done, thank you.
  • Page 10, line 391: “to generate” à to be regenerated We replaced “generate” with “establish”.
  • Page 10, line 392: “major limitations are” à there are still many limitations including We rephrased the sentence as you suggested (page 10, line 416/417).
  • Page 10, line 396: “a post entry block” à a post entry blockade Done, thank you.
  • Page 10, line 408: “does still” à still does Done, thank you.
  • Page 10, line 411: “However to …“ à However, to mimic a chronic and long-lasting HDV infection as observed in patients, we need to identify and humanize further host restriction factors … Done, thank you.
  • Page 13, line 493: “with” à by Done, thank you.

Reviewer 2 Report

Review on virus-1161279

The authors described current problems on hepatitis D virus (HDV) infection systems. Such are also problems on hepatitis B virus (HBV) infection system, since HDV infects cells through HBV envelope proteins as you know.

The context is well-written and probably good enough for publication as it is. I, however, would like to ask the authors to discuss difference in infection of HDV and HBV to mouse cells expressing human NTCP (hNTCP), since HDV can infect to hNTCP expressing mice hepatocytes though with very low efficiency. On the other hand, HBV never infects the cells.

Specific point:

  1. Line 183, “Guilhot et al” should be “Guilhot et al.”.

Author Response

Reviewer 2: Comments and Suggestions for Authors

Review on virus-1161279

The authors described current problems on hepatitis D virus (HDV) infection systems. Such are also problems on hepatitis B virus (HBV) infection system, since HDV infects cells through HBV envelope proteins as you know.

The context is well-written and probably good enough for publication as it is. I, however, would like to ask the authors to discuss difference in infection of HDV and HBV to mouse cells expressing human NTCP (hNTCP), since HDV can infect to hNTCP expressing mice hepatocytes though with very low efficiency. On the other hand, HBV never infects the cells.

Thank you for your comments and suggestion.

We addressed the point made by the reviewer and mentioned differences of HBV and HDV to infect murine cells and discussed why murine hepatocytes might not support HBV infection at page 11. Since HBV and HDV share the same HBV envelope proteins, entry of HBV and HDV is supposed to be similar. However, the HBV post entry steps, which finally lead to the formation of the cccDNA as a transcriptional template for HBV, are currently not fully understood but they appear complex and require several host proteins, which might be missing in murine cells or are not fully capable to support cccDNA formation in murine hepatocytes. This hypothesis is supported by the notion that HBV transgenic mice support viral replication but not cccDNA formation, while transfection of a cccDNA-like molecule into non-susceptible murine cells supports HBV antigen expression.

Specific point:

  1. Line 183, “Guilhot et al” should be “Guilhot et al.”. Done, thank you.
